# Multiplex secretome engineering enhances recombinant protein production and purity

Stefan Kol [1✉], Daniel Ley[1], Tune Wulff[1], Marianne Decker[1], Johnny Arnsdorf[1], Sanne Schoffelen [1], Anders Holmgaard Hansen [1], Tanja Lyholm Jensen[1], Jahir M. Gutierrez[2,3], Austin W. T. Chiang[2,4], Helen O. Masson[2,3], Bernhard O. Palsson[1,2,3,4], Bjørn G. Voldborg[1], Lasse Ebdrup Pedersen[1], Helene Faustrup Kildegaard[1], Gyun Min Lee[1,5] & Nathan E. Lewis[2,3,4✉]

Host cell proteins (HCPs) are process-related impurities generated during biotherapeutic protein production. HCPs can be problematic if they pose a significant metabolic demand, degrade product quality, or contaminate the final product. Here, we present an effort to create a "clean" Chinese hamster ovary (CHO) cell by disrupting multiple genes to eliminate HCPs. Using a model of CHO cell protein secretion, we predict that the elimination of unnecessary HCPs could have a non-negligible impact on protein production. We analyze the HCP content of 6-protein, 11-protein, and 14-protein knockout clones. These cell lines exhibit a substantial reduction in total HCP content (40%-70%). We also observe higher productivity and improved growth characteristics in specific clones. The reduced HCP content facilitates purification of a monoclonal antibody. Thus, substantial improvements can be made in protein titer and purity through large-scale HCP deletion, providing an avenue to increased quality and affordability of high-value biopharmaceuticals.

[1] The Novo Nordisk Foundation Center for Biosustainability, Technical University of Denmark, Building 220, Kemitorvet, 2800 Kgs. Lyngby, Denmark. [2] The Novo Nordisk Foundation Center for Biosustainability at the University of California, San Diego, School of Medicine, La Jolla, CA 92093, USA. [3] Department of Bioengineering, University of California, San Diego, School of Medicine, La Jolla, CA 92093, USA. [4] Department of Pediatrics, University of California, San Diego, School of Medicine, La Jolla, CA 92093, USA. [5] Department of Biological Sciences, KAIST, 291 Daehak-ro, Yuseong-gu, Daejeon 305-701, Republic of Korea. ✉email: stef.kol@gmail.com; nlewisres@ucsd.edu

Host cell proteins (HCPs) that are released from dead cells and secreted from viable cells accumulate extracellularly during mammalian cell culture, potentially impairing product quality[1,2] and posing an immunogenic risk factor[3]. HCPs must be reduced to low levels (<1–100 ppm) in all cell-derived protein biotherapeutics before final product formulation, putting a great demand on downstream purification. Depending on titer, downstream processing comprises up to 80% of the entire production costs of a monoclonal antibody (mAb)[4]. In the production of a mAb, protein A affinity capture is often employed as a generic first step, followed by two or three orthogonal polishing steps[5]. Although the concentrations of HCPs in cell culture harvests are reduced to acceptable levels after these purification steps, certain HCPs can escape an entire purification process and remain in the final product at levels that may affect product quality and stability, cause an immunogenic reaction, or display residual activity[6,7]. It is therefore of utmost importance to closely monitor and identify HCPs to ensure their removal from biotherapeutic products. Enzyme-linked immunosorbent assay (ELISA) is typically used in industrial processes to monitor the total HCP content, but gives no information on specific HCP components[8]. Proteomic analysis, such as two-dimensional gel electrophoresis and/or mass spectrometry (MS), can therefore be employed to identify specific HCPs[6,9,10]. Although many studies have focused on identification, only a few efforts have been made to remove troublesome HCP moieties by host cell line engineering[1,2,11,12].

Concerning CHO HCPs that affect product quality, a matriptase-1 knockout cell line was recently created that prevents degradation of recombinant proteins[2]. A knockout cell line has also been created devoid of lipoprotein lipase (Lpl) activity[1]. Lpl is a difficult-to-remove HCP that degrades polysorbate in mAb formulations, and targeted mutation resulted in abolishment of Lpl expression and reduced polysorbate degradation without substantial impact on cell viability. In a recent study, the HCPs Annexin A2 and cathepsin D were removed[12]. Although the authors do not confirm loss of activity, the knockouts do not adversely affect cell growth. In addition, knockout of a serine protease was shown to eliminate proteolytic activity against a recombinantly expressed viral protein[11]. These previous efforts targeted single genes, but mammalian cells can secrete a few thousand HCPs[9], so CRISPR technology could be used to target multiple genes simultaneously by simply expressing multiple sgRNAs together with a single Cas9 nuclease[13,14].

Here, we demonstrate substantial reductions in HCP content can be obtained through targeted genome editing, guided by omics analyses. Specifically, we used a systems biology computational model to show that the removal of multiple HCPs could free up a non-negligible amount of energy. We then used proteomics to help identify target HCPs and subsequently applied multiplex CRISPR-Cas9 to disrupt up to 14 genes coding for proteins that are abundant in harvested cell culture fluid (HCCF), difficult-to-remove during downstream processing, or have a potentially negative impact on product quality. We analyzed HCP content of a 6-protein, 11-protein, and 14-protein knockout and characterized their growth in shake flasks, Ambr bioreactors, and DASGIP bioreactors. We observed a substantial reduction of total HCP content in the 6xKO (~40%), 11xKO, and 14xKO (60–70%) cell lines. Depending on the cell line and growth conditions, we also observed increased mAb productivity and improved growth characteristics. When measuring HCP content and mAb titer at the different stages of a three-step purification process (using protein A and two ion exchange chromatography steps), we observed a strong reduction in HCP content, increased HCP log reduction values (LRV) of protein A affinity chromatography, and decreased HCP ppm values. Thus, we demonstrate that

superior CHO production host cell lines can be produced by eliminating specific HCPs. The dramatically lowered HCP ppm values during purification of a mAb will facilitate downstream processes. These HCP-reduced knockouts can be combined with additional advantageous genetic modifications, such as the production of glycoengineered proteins[15–18], higher viability during long culture times[19], viral resistance or elimination[20–22], and/or clones with higher protein production stability of mAbs[23], to create predictable upstream and downstream processes with full control over critical process parameters.

## Results

**CHO cells spend substantial resources on the secretome**. There are several potential advantages to the removal of HCPs, including the elimination of immunogenic proteins or proteins that degrade product quality. However, the question remains regarding the bioenergetic investment made by the host cells on the secreted fraction of their proteome. To evaluate the resource and bioenergetic investment of CHO cells into their secretome, we calculated the cost of each protein using a model of CHO cell protein secretion[24], scaled by its expression in RNA-Seq data[25]. In CHO-S cells, we found that 39.2% of the resources are dedicated to producing membrane and secreted proteins, most of which are processed through the secretory pathway (Fig. 1a). In all, 10.2% of cell resources were explicitly dedicated to secreted proteins with signal peptides, and a relatively small number of secreted proteins account for the majority of the bioenergetic cost and use of the secretory pathway. Furthermore, we anticipate that the cost for secreted proteins is likely higher since these proteins are not turned over and recycled like intracellular and membrane proteins. Thus, there is great potential to free up cellular resources and secretory capacity, in addition to improving product quality by eliminating unnecessary HCPs.

**Target selection and verification**. To decrease contaminating HCP secretion and free up resources for CHO cell growth and protein production, we identified targets based on three criteria: (i) being abundant in CHO supernatants as analyzed by mass spectrometry (Supplementary Data 1), (ii) being a difficult-to-remove impurity in downstream processing[26–29], and (iii) having a negative impact on product quality[1,30,31]. To remove the genes, we subjected the cell lines in this study to 4 cycles of CRISPR-Cas9-mediated multiplex gene disruption. In the first round, the gene encoding Metalloproteinase inhibitor 1 (Timp1) was disrupted. In the second round, the genes encoding Biglycan (BGN), Galectin-3-binding protein (LGALS3BP), Nidogen-1 (NID1.1 and NID1.2), and Cathepsin D (CTSD) were disrupted, resulting in the 6xKO cell line. In the third round, the genes encoding *N* (4)-(Beta-*N*-acetylglucosaminyl)-L-asparaginase (Aga), Endoplasmic reticulum resident protein 29 (Erp29), G-protein coupled receptor 56 (Gpr56), Tubulointerstitial nephritis antigen-like (Tinagl1), and Legumain (Lgmn) were disrupted, resulting in the 11xKO cell line. Finally, in the fourth round, we disrupted the genes encoding YEATS domain-containing protein 2 (Yeats2), SPARC (Sparc), and Lipoprotein lipase (Lpl), resulting in the 14xKO cell line.

The targets were predominantly identified through proteomics on spent media or purified product, and we verified that these genes were, on average among the more abundant transcripts for secreted proteins, and accounted for proteins exerting a greater cost to the cell (Fig. 1b). Furthermore, upon quantifying the amount of ATP liberated upon their deletion, we found their deletion led to a significantly higher amount of resources eliminated ($p \ll 1 \times 10^{-5}$, randomization test), when compared to a comparable number of randomly selected genes.

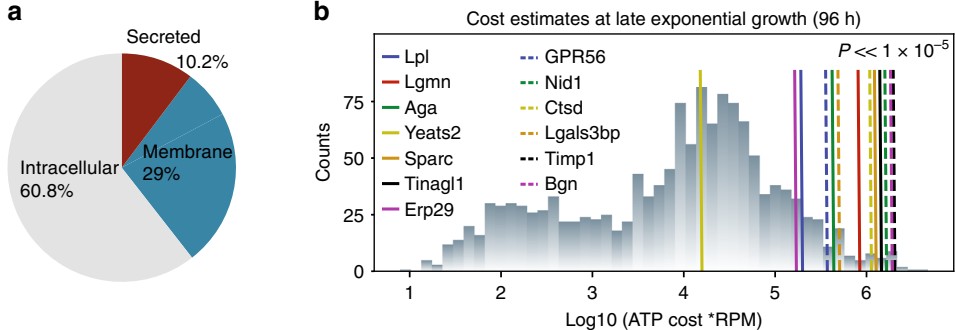

**Fig. 1 Substantial resources can be liberated by targeting HCPs in the CHO secretome. a** Proteins produced through the secretory pathway account for more than 40% of the cell's biosynthetic capacity, and roughly 1/4 of this cost is dedicated to secreted proteins. **b** Following proteomic analysis of the HCCF, 14 proteins (associated with 13 gene loci) were identified and targeted for deletion. The bioenergetic cost of protein synthesis and secretion of these target genes was quantified as ATP cost per reads mapping to gene per million reads in the transcriptome (RPM). Deletion will free up significantly more bioenergetic resources than expected from the deletion of a similar number of randomly selected genes ($p \ll 1 \times 10^{-5}$). This randomization test was a one-sided test and, as it is not a multiple comparison, no adjustment was made. Source data are provided as a Source Data file.

**Table 1 Selection of targets and MiSeq verification of knockouts.**

| Cell line | Target | Protein name | Uniprot | Rationale | Location | Frameshift |
|---|---|---|---|---|---|---|
| 6xKO | BGN | Biglycan | G3HSX8 | A | Secreted | +1 |
| | TIMP1 | Metalloproteinase inhibitor 1 | G3IBH0 | A/C | Secreted | −5/−2/+1 |
| | LGALS3BP | Galectin-3-binding protein | G3H3E4 | A/C | Membrane | +33 |
| | NID1.1 | Nidogen-1 | G3HWE4 | A/C | Membrane/matrix | +1 |
| | NID1.2 | Nidogen-1 | G3I3U5 | A/C | Membrane/matrix | +1 |
| | CTSD | Cathepsin D | G3I4W7 | A/C/Q | Lysosome | −8 |
| 11xKO | Aga | N(4)-(Beta-N-acetylglucosaminyl)-L-asparaginase | G3HGM6 | A | Secreted | −40 |
| | Erp29 | Endoplasmic reticulum resident protein 29 | G3H284 | A | ER | −1 |
| | Gpr56 | G-protein coupled receptor 56 | G3I3K5 | A/C | Membrane | +1 |
| | Tinagl1 | Tubulointerstitial nephritis antigen-like | G3H1W4 | A/C | Secreted | +1 |
| | Lgmn | Legumain | G3I1H5 | A/C/Q | Lysosome | +1 |
| 14xKO | Yeats2 | YEATS domain-containing protein 2 | G3I3I8 | A | Nucleus | +1 |
| | Sparc | SPARC | G3H584 | A/C | Secreted | −5 |
| | Lpl | Lipoprotein lipase | G3H6V7 | A/C/Q | Membrane/secreted | +1 |

The target names, full protein names, UniProt identifiers, rationale for deletion, subcellular location, and indel size are indicated. Source data are provided as a Source Data file.
A abundance, C copurifying, Q quality.

We verified gene disruption at the genomic and proteomic levels. Specifically, we used MiSeq analysis (Table 1) to verify that all indels led to frameshift mutations except for the one generated in LGALS3BP, which leads to a partly scrambled amino acid sequence and the introduction of a stop codon after residue 44. Mass spectrometry was subsequently used to verify the absence of our target proteins after gene disruption. Supernatants from wild-type CHO-S (WT) and the 6xKO, 11xKO, and 14xKO cell lines were analyzed for the presence of peptides derived from the target proteins (Fig. 2a–c). Except for the target YEATS2, peptides were detected for all of the targets in WT. No or very few peptides were detected in the 6xKO, 11xKO, and 14xKO cell lines as compared to WT for all targets except LGALS3BP. Approximately 10–20% of LGAL3SBP peptides derived from the region upstream of the editing site were still present in all knockout cell lines, indicating secretory downregulation of the partial non-sense protein produced by the frameshift mutation. All other targets were no longer detected.

**KOs display improved growth and less HCP content.** To assess the properties of our knockout cell lines, we followed their growth and viability for 7 days in shake flasks with daily sampling to measure viable cell density (VCD), cell viability, and HCP content

(Fig. 3). Interestingly, cell density and viability of knockout cell lines were improved over WT (Fig. 3a). While the 6xKO cell line displayed an intermediate phenotype, growth of the 11xKO and 14xKO cell lines was enhanced to reach a VCD of ~7.6 million cells per milliliter. WT reached a VCD of ~5.4 million cells per milliliter. Viability of WT started to decrease on the fifth day of culture, while the knockout cell lines remained above 90% viability for the duration of the experiment. Samples were analyzed using a CHO HCP detection kit and showed a remarkable reduction in HCP content as compared to WT (Fig. 3b). Whereas the 6xKO cell line shows an intermediate phenotype again, the 11xKO and 14xKO cell lines behave similarly, showing an HCP reduction to ~150 µg mL$^{-1}$. As the VCD of the knockout cell lines is higher, the behavior of the cell lines can arguably better be described by calculating specific HCP productivities. Specific HCP productivity was found to be 13.5 picograms per cell per day (pcd) for WT, whereas the 6xKO, 11xKO, and 14xKO cell lines were reduced to 9.9, 5.7 and 5.6 pcd, respectively (Fig. 3c). Specific HCP productivity was found to be reduced by 27%, 58% and 59% for the 6xKO, 11xKO, and 14xKO cell lines, respectively, as compared to WT (Fig. 3d). Besides the improvements in growth characteristics during batch cultivation, we encountered an unexpectedly high reduction in HCP content. As it could be

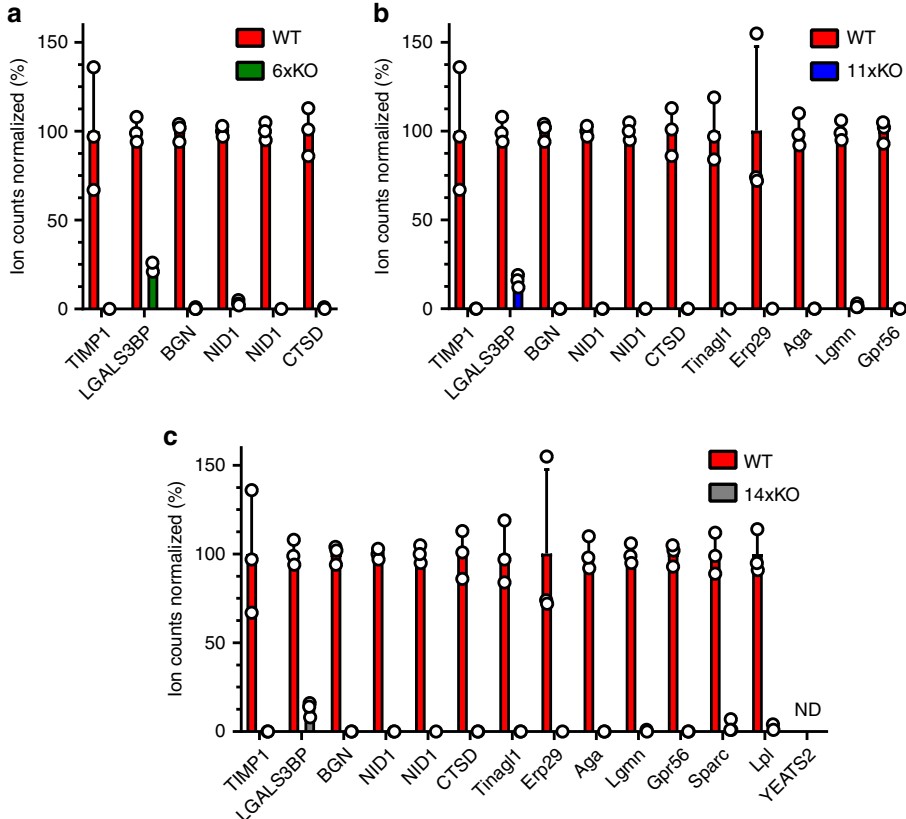

**Fig. 2 Verification of knockouts by mass spectrometry.** Supernatant samples prepared from WT (red bars) were subjected to MS and compared to the **a** 6xKO (green bars), **b** 11xKO (blue bars) and **c** 14xKO (gray bars) cell lines with respect to the indicated targets. Ion counts were normalized to the levels detected in WT. The target YEATS2 was not detected (ND) in both WT and the 14xKO cell line. All data are presented as mean values ± SD ($n = 3$ independent experiments). Source data are provided as a Source Data file.

argued that the HCP-reduced phenotype is an artifact of the knockout procedure, we determined the HCP content of unrelated cell lines harbouring multiple knockouts generated using the same procedure. No significant decrease in HCP content of WT CHO-S and the knockout cell lines is observed (Supplementary Fig. 1), showing that the HCP-reduced phenotype is specifically caused by the removal of our targeted proteins. To gain a better understanding of the impact of these knockouts, we measured several basic cell properties and established whether mAb productivity was affected.

**KOs display higher productivity and less HCCF total protein.** To assess several basic cell properties and protein productivity, we generated stable pools expressing the monoclonal antibody Rituximab (Rit). We cultivated the WT, WT Rit, 6xKO Rit, 11xKO Rit, and 14xKO Rit cell lines for 4 days in shake flasks and collected HCCF and cell pellets. We analyzed HCP content, protein content of HCCF and cells, cell size, transfection efficiency, antibody staining efficiency, and Rituximab titer (Fig. 4 and Supplementary Fig. 2). The HCP-reduced phenotype was observed as before (Supplementary Fig. 2b), while VCD was similar for all cell lines (Supplementary Fig. 2a). The total protein content of the HCCF obtained from the knockout cell lines was significantly reduced by ~50% (Fig. 4a), while cell size was reduced from 13.4 μm to 12.5 μm (Supplementary Fig. 2c). The protein content per cell remained unchanged at ~150-pg protein per cell (Fig. 4b), which corresponds well with published values using a CHO-K1 cell line[32]. Antibody productivity in cell pools was assayed in two ways: (1) by an antibody staining method and subsequent FACS analysis (Fig. 4c) and (2) by biolayer

interferometry (Fig. 4d). The knockout cell lines displayed higher productivity as compared to WT using both methods. Whereas productivity increased with the number of knockouts using the staining method, it is only increased until the 11xKO cell line using biolayer interferometry. Transfection efficiency was similar for all cell lines (Supplementary Fig. 2d). Based on these results, we selected high-producing clones to analyze their growth properties under fed-batch conditions. During selection, the 14xKO Rit cell line displayed a low single-cell survival rate and no clones could be selected that displayed sufficiently high mAb productivity. Based on these observations and as the HCP-reduced phenotype was not decreased further in the 14xKO cell line, we excluded the 14xKO Rit cell line from further study.

**KOs display decreased HCP content in fed-batch cultivation.** After selecting high mAb-producing clones of the WT Rit, 6xKO Rit, and 11xKO Rit cell lines, we analyzed their growth properties, productivity, and HCP content in Ambr (Supplementary Fig. 3) and DASGIP bioreactors (Fig. 5). The goal of these experiments was to compare behavior of clones in different media and to generate material for downstream processing. In addition, we wanted to ensure that high producers could be generated using the knockout cell lines and that these retained the HCP-reduced phenotype under controlled cultivation conditions.

Media screening using the Ambr bioreactor revealed that high-producing mAb clones could be selected from all remaining knockout mutants (Supplementary Fig. 3). As productivity was most comparable across mutants in FortiCHO medium, we selected this medium to scale up cultivation in a DASGIP bioreactor (Fig. 5). To limit the contribution of lysed cells to HCP

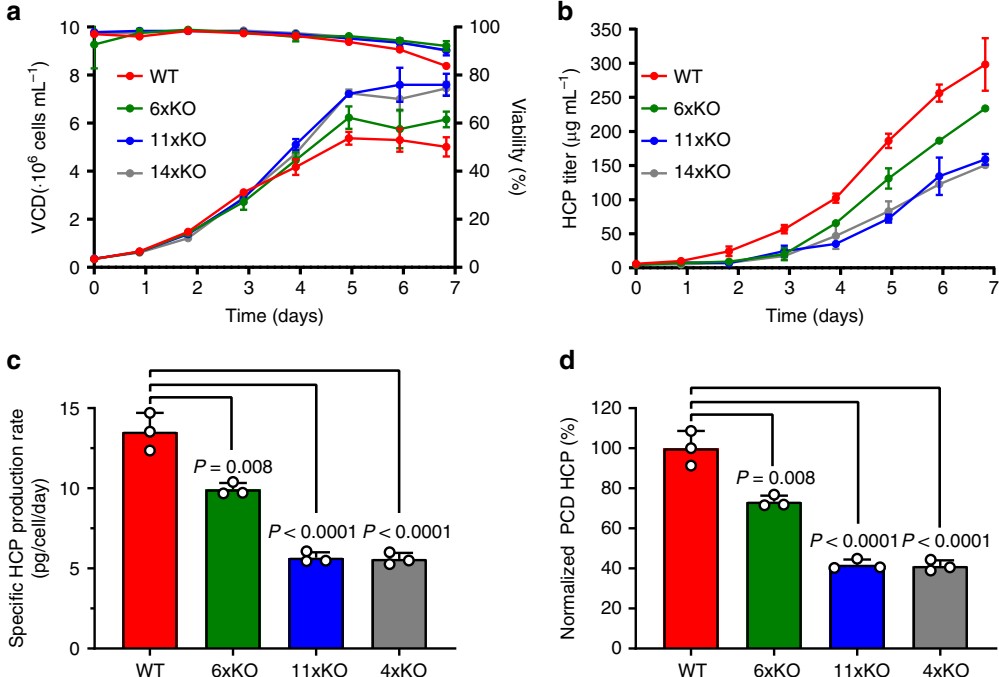

**Fig. 3 Knockout cell lines display improved growth and HCP profile. a** Viability and VCD, **b** HCP profile, **c** normalized HCP content on day 7, and **d** normalized specific HCP productivity of WT (red), 6xKO (green), 11xKO (blue), and 14xKO (gray) cell lines. Viability, VCD and HCP content were measured from day 0 to day 7 in shake flasks. The specific HCP productivity was determined as the slope of HCP concentration versus the integral number of viable cell (IVCD) calculated from day 0 to day 7, and expressed as pg HCP per cell and per day (HCP pcd). Specific HCP production was normalized to the levels detected in WT. Data are presented as mean values ± SD ($n = 3$ independent experiments). Statistical analysis (**c**, **d**) was performed using one-way ANOVA followed by Tukey's post-hoc test. Source data are provided as a Source Data file.

content in the downstream process, we harvested when the first culture dropped below 90% viability (Fig. 5a, arrow). Maximum VCD of WT Rit and the 11xKO Rit cell lines were comparable (Fig. 5a), while the VCD of the 6xKO Rit cell line was ~50% lower (also observed in the Ambr bioreactor). However, Rituximab titer at harvest (Fig. 5b) of the 6xKO Rit cell line was comparable to WT Rit, while the 11xKO Rit cell line produced ~40% less. HCP titer reduction of the 6xKO Rit and 11xKO Rit cell lines was even more pronounced under these circumstances. Whereas the WT Rit cell line produces 1080 µg mL$^{-1}$ HCP, the 6xKO and 11xKO cell lines behave similarly and produce ~200 µg mL$^{-1}$ HCP (Fig. 5c). Specific HCP productivity was 19.7 pcd for WT Rit, whereas the 6xKO Rit and 11xKO Rit cell lines were reduced to 5.9 and 4.8 pcd, respectively (Fig. 5d). To emphasize the substantial improvement in product purity, we calculated the mAb/HCP ratio, which is improved 7-fold in the 6xKO Rit cell line and 2-fold in the 11xKO cell line (Fig. 5e). Additional clones also displayed improved mAb/HCP ratios (Supplementary Fig. 4), showing that the observed phenotype is not caused by clonal variation. Importantly, these experiments show that, under controlled cultivation, our knockout cell lines retain the strongly HCP-reduced phenotype and improved product purity. In addition, as the selection process of mAb producers required more than 56 days in culture, the observed phenotype was stable for at least 64 generations.

**mAb purification results in a pure and bioactive product**. To assess the effect of the knockouts on downstream purification, we performed protein A affinity chromatography, followed by cation exchange chromatography (CIEX) and anion exchange chromatography (AIEX) (Supplementary Fig. 5). Samples were collected after every chromatographic step and were analyzed with respect to mAb concentration and HCP content. Using these data, we

calculated mAb purification efficiency, HCP content per mg mAb in ppm, and HCP log reduction values (LRV) (Table 2). Purification efficiency ranges from 35% to 55% and total HCP LRV of the process ranges from 4.4 to 5.8. HCP content after three chromatographic steps was 45 ppm, 18 ppm, and 3 ppm for the WT Rit, 6xKO cell, and 11xKO cell lines, respectively. Besides the decrease in HCP content in the 11xKO cell line throughout mAb purification, we also observed an increase in HCP clearance during protein A chromatography of one order of magnitude.

To ensure that the mAb product quality was not affected by the HCP knockouts, the Rituximab product from the mutant clones was subsequently subjected to glycosylation analysis[33,34] (Supplementary Fig. 6a) and in vitro binding to a cell line expressing CD20, the cellular target of Rituximab (Supplementary Figs. 6b and 7). The glycosylation profile and bioactivity of Rituximab produced in the knockout cell lines were found to be very similar to wild-type produced Rituximab. The stability of mAb glycosylation was also supported by RNA-Seq analysis of the WT, 6XKO, and 11XKO cell lines, wherein glycosylation was not significantly perturbed (Supplementary Tables 1 and 2 and Supplementary Figs. 8–10). We conclude that the HCP-reduced phenotype is maintained throughout a representative downstream process and that the bioactivity of the product is not perturbed.

## Discussion

The yield of recombinant proteins during biomanufacturing has steadily increased over the past decades. Thus, downstream processing has become a bottleneck in biotherapeutic production[35]. Efforts to alleviate this have mostly focused on innovations in process design to improve downstream processing capacity, speed, and economics[36,37]. It has also been suggested that cell line engineering could be used to reduce or remove problematic

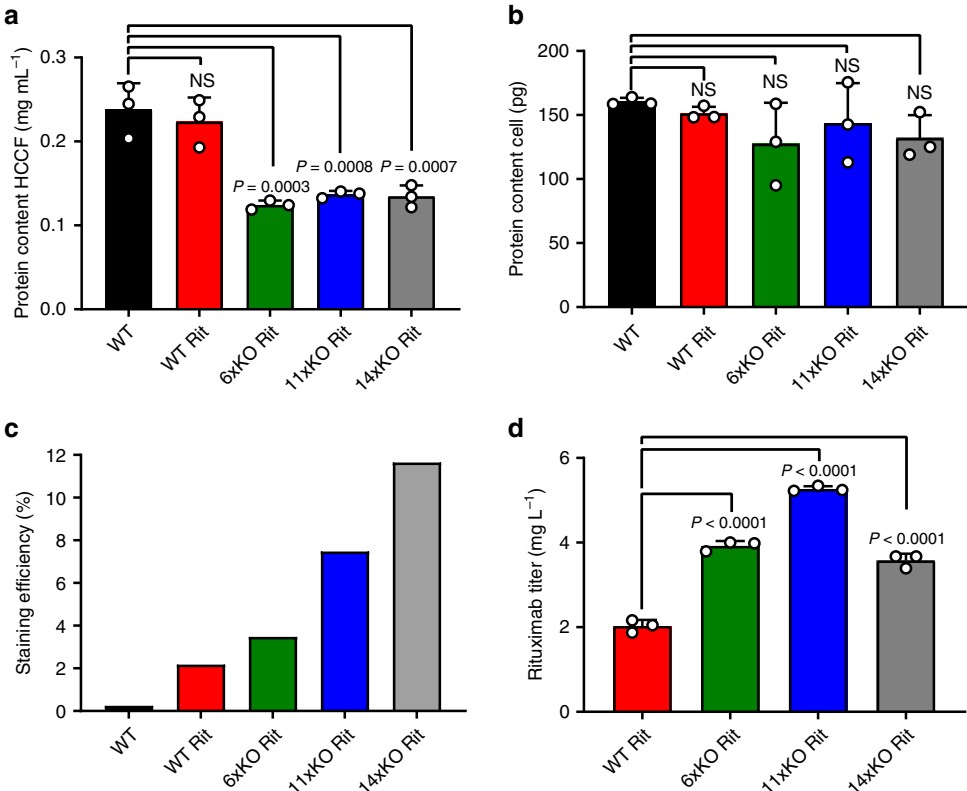

**Fig. 4 Knockout cell lines display increased productivity and less HCCF protein content. a** HCCF protein content, **b** protein content per cell, **c** staining efficiency, and **d** Rituximab titer of WT (black), WT Rit (red), 6xKO Rit (green), 11xKO Rit (blue), and 14xKO Rit (gray) cell lines. All properties were measured after cultivation for 4 days in shake flasks. Data are presented as mean values ± SD ($n = 3$ independent experiments) except for the staining efficiency determination. Statistical analysis (**a**, **b**, **d**) was performed using one-way ANOVA followed by Tukey's post-hoc test (ns = not significant). Source data are provided as a Source Data file.

HCPs, lessening the need for additional purification steps[6,38]. Indeed, researchers have specifically removed unwanted proteins from CHO cell lines that affect product quality[1,2,11,12]. In addition, it has recently been shown that depletion of a highly abundant mRNA encoding a non-essential protein can improve growth and product titer, indicating that it is possible to free up energetic and secretory resources by rational cell line engineering strategies[39]. This observation was supported by a model of the secretory pathway, which was able to accurately predict the increase in growth and protein productivity[24]. These studies have so far been limited to single engineering targets, but could easily be applied on a larger scale. Although different in approach, a recent paper pioneers the use of inducible downregulation of non-product related protein expression leading to an increase in the purity of the biopharmaceutical product[40]. Knowledge-based decisions can support purity-by-design approaches and could lead to easier downstream purification processing.

Here, we hypothesized that removal of HCPs by targeted gene disruption will lead to improvements in cell growth, secretory pathway capacity, and DSP. We used computational modeling to demonstrate that CHO cells spend substantial resources on the host cell secretome. We proceeded with the removal of 14 HCPs, which were more highly abundant in CHO HCCF. We also prioritized targets that can be difficult-to-remove or that influence product quality. Surprisingly we found the removal of these 14 HCPs led to a substantial decrease in HCP content of up to 70% and HCCF total protein content of up to 50%. The observed phenotype persists under controlled cultivation and is specifically caused by the knockouts. The HCP-reduced phenotype was stable over many generations and was observed in multiple clones. The

ability to generate high mAb producers was not perturbed in the 6xKO and 11xKO cell lines and the resulting mAb product was shown to be indistinguishable from a mAb produced in a wild-type CHO cell line.

Improvements in product purity will considerably facilitate the purification of mAbs to acceptable HCP levels, especially in those situations where HCPs form a challenge in the purification or quality of a biotherapeutic protein. It is tempting to speculate that one of the ion exchange chromatography steps could be omitted from the downstream purification process. However, besides the removal of HCP impurities, ion exchange chromatography is also used to reduce high molecular weight aggregates, charge-variants, residual DNA, leached Protein A, and viral particles[41]. It remains to be seen whether two chromatographic steps can bring other product- and process-related impurities within regulatory requirements. The results presented here certainly warrant a detailed analysis of the HCP-reduced cell lines under industrial mAb production conditions with respect to growth, production, and mAb critical quality attributes.

Systems biology approaches and omics data analysis provide useful tools for screening and prioritizing HCPs that are more likely to free up greater amounts of cellular resources upon removal[24,42]. Through a multiplex genome editing approach, we successfully removed more than half of the HCPs in CHO cells by mass. This demonstration highlights that large-scale genome editing in mammalian cells can be done for bioproduction, and can have benefits throughout the bioproduction process. In particular, such efforts can be effectively used to make clean CHO cells by eliminating many undesirable HCPs[6] and retroviral-like particles[22,43] that currently require extensive purification

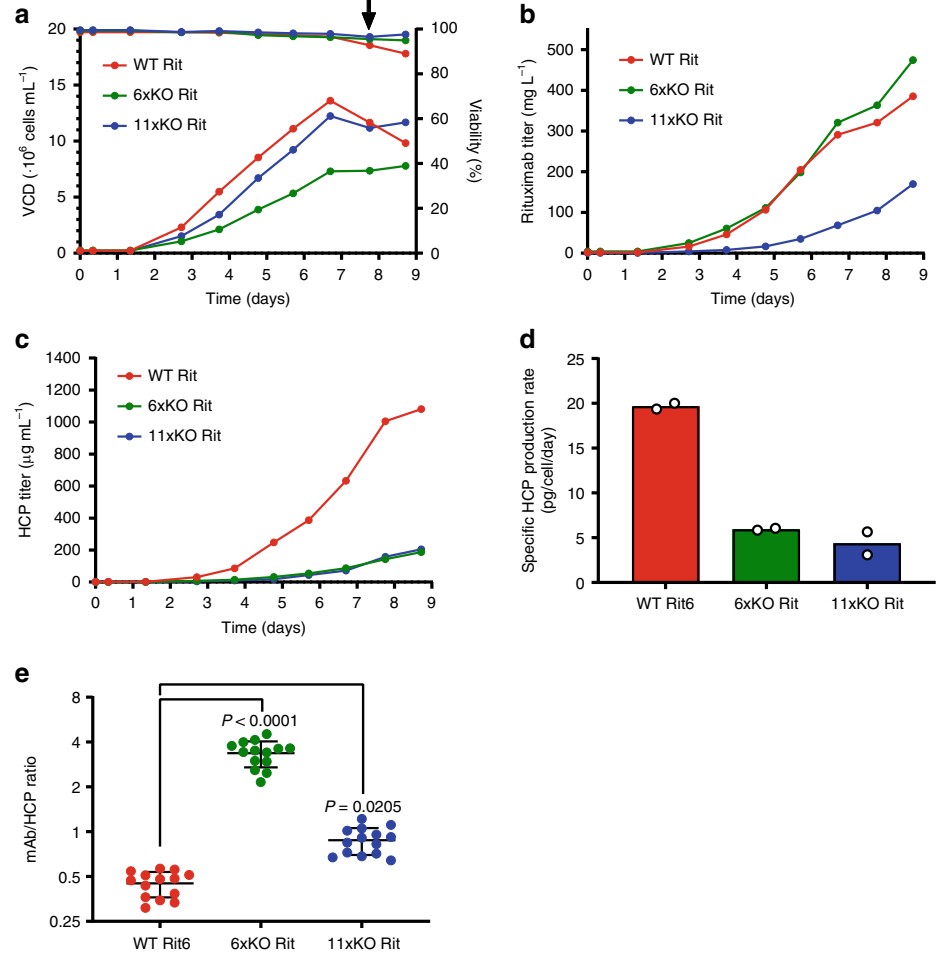

**Fig. 5 The HCP-reduced phenotype is maintained under fed-batch cultivation. a** Viability and VCD, **b** Rituximab titer, **c** HCP profile, **d** specific HCP productivity, and **e** the mAb/HCP ratio of WT Rit (red), 6xKO Rit (green), and 11xKO Rit (blue) cell lines. Each cell line was cultivated in duplicate. Viability, VCD, titer, and HCP content were measured daily in a DASGIP bioreactor. Mean of technical duplicates is shown. At the indicated timepoint (arrow), 100 mL of the culture was harvested for mAb purification. The HCP/mAb ratio was calculated by dividing the mAb titer by the HCP titer for all 7 time points in duplicate. Data (**e**) is presented as mean values ± SD ($n = 14$ independent samples). Statistical analysis was performed using one-way ANOVA followed by Tukey's post-hoc test. Source data are provided as a Source Data file.

**Table 2 Knockout cell lines display reduced HCP content during a purification process.**

|  |  | HCCF | Protein A | CIEX | AIEX | Total |
|---|---|---|---|---|---|---|
| WT | mAb (µg) | 38,470 | 33,135 | 26,371 | 17,088 |  |
|  | Purification yield (%) |  | 86.1 | 79.6 | 64.8 | 44.4 |
|  | HCP (ppm) | 1,795,217 | 7243 | 158 | 45 |  |
|  | LRV |  | 2.4 | 1.7 | 0.5 | 4.6 |
| 6xKO | mAb (µg) | 43,480 | 41,891 | 31,904 | 24,104 |  |
|  | Purification yield (%) |  | 96.3 | 76.2 | 75.6 | 55.4 |
|  | HCP (ppm) | 472,873 | 1596 | 74 | 18 |  |
|  | LRV |  | 2.5 | 1.3 | 0.6 | 4.4 |
| 11xKO | mAb (µg) | 12,180 | 10,881 | 7377 | 4308 |  |
|  | Purification yield (%) |  | 89.3 | 67.8 | 58.4 | 35.4 |
|  | HCP (ppm) | 1,638,998 | 551 | 47 | 3 |  |
|  | LRV |  | 3.5 | 1.1 | 1.2 | 5.8 |

The HCCF, protein A elution pool, CIEX elution pool, and AIEX flow through pool of WT Rit, 6xKO Rit, and 11xKO Rit were subjected to mAb and HCP titer measurements. These values were used to calculate yield, HCP LRV, and HCP content in ppm. Mean of technical duplicates is shown.
*CIEX* cation exchange, *AIEX* anion exchange, *LRV* log reduction value, *ppm* parts per million.

strategies. Through this, one may obtain higher quality drugs and produce these at lower costs.

## Methods

**Cell culture and cell line generation.** All cell lines described here were created using CHO–S cells (Life Technologies, Carlsbad, CA, USA). Knockout cell lines were generated by transfecting with expression vectors encoding GFP_2A_Cas9 and sgRNAs[14]. Fluorescent cells were single-cell sorted by FACS on day 2 after transfection. Primers used in this study are listed in Supplementary Data 2. Viability, cell size and VCD was monitored using the NucleoCounter NC-200 Cell Counter (ChemoMetec, Allerod, Denmark). Batch cultivation was performed in 250 mL shake flasks containing 60–80 mL CD-CHO medium (Thermo Fisher Scientific, Waltham, MA, USA) supplemented with 8 mM L-glutamine (Life Technologies) and 0.2% anti-clumping reagent (Life Technologies). Cells were extracted from the cell bank and thawed in 10-mL pre-warmed medium. Cultures were passaged three times during 7 days and strained through a 40-µm filter before inoculation. Cultivation was performed in a humidified shaking incubator operated at 37 °C, 5% $CO_2$ and 120 rpm.

Prior to inoculation into fed-batch bioreactors, cells were adapted from CD-CHO medium to FortiCHO (Gibco, Carlsbad, CA, USA), OptiCHO (Gibco), or ActiPro (GE Healthcare, Chicago, IL, USA) medium during three passages. All media were supplemented with 1% Anti-Anti (Life Technologies), 0.1% anti-clumping agent (Life Technologies), and 8 mM L-glutamine (Lonza, Basel, Switzerland). Fed-batch cultivation was performed in a DASGIP Mini bioreactor (Eppendorf, Jülich, Germany) containing between 260 and 280 mL medium or in an Ambr 15 bioreactor system (Sartorius, Göttingen, Germany) containing 14.4 mL medium. Temperature was maintained at 37 °C. In the DASGIP, dissolved oxygen (DO) was maintained at 40% by cascade aeration strategy (Air/Air&$O_2$/$O_2$) using

nitrogen as a carrier gas (total flow rate 1–3 L h$^{-1}$). In the Ambr, DO was maintained at 40% using pure O$_2$ using nitrogen as carrier gas (total flow rate 0.200 sL h$^{-1}$). Agitation rate (200–400 rpm) was adjusted on a daily basis to keep DO set point at 40%. Culture pH was maintained at 7.1 ± 0.02 (DASGIP) or 7.1 ± 0.05 (Ambr) using intermittent addition of CO$_2$ to the gas mix and 1 M CHNaO$_3$. HyClone Cell Boost 7a supplement (GE Healthcare) containing 500 mM glucose and HyClone Cell Boost 7b Supplement (GE Healthcare) were fed daily when cell numbers reached 1.4–2.2 × 10$^6$ cells mL$^{-1}$ at 2% and 0.2% of daily total volume, respectively. Glucose (2220 mM) and glutamine (200 mM) were used to maintain levels between 24 and 30 mM, and 3–5 mM, respectively, via a once-daily feeding based on measured concentration, doubling time, and consumption rates. Antifoam C (Sigma, St. Louis, MI, USA) was supplemented when needed.

**Deep sequencing analysis of genome edit sites**. Confluent colonies from 96-well flat-bottom replicate plates were harvested for genomic DNA extraction. DNA extraction was performed using QuickExtract DNA extraction solution (Epicentre, Madison, WI, USA) according to the manufacturer's instruction. The library preparation was based on Illumina 16S Metagenomic Sequencing Library. Preparation and deep sequencing were carried out on a MiSeq Benchtop Sequencer (Illumina, San Diego, CA). The protocol for amplifying the CRISPR-targeted genomic sequences, amplicon purification, adapter-PCR, and quality analysis followed our previously published methods[14].

**Generation of antibody expressing cell lines**. WT and knockout cell lines (6xKO, 11xKO, and 14xKO) were initiated from cryopreserved vials and passaged for 1 week prior to transfection. Transfection efficiency was determined by transfecting the pMAX-GFP (Lonza) and subsequent FACS analysis. Cells were transfected with a plasmid encoding the antibody Rituximab and the zeocin selection gene using Freestyle MAX reagent and maintained for 2 days before selection pressure was initiated with zeocin (400 µg mL$^{-1}$). After 21 days of selection, viability was above 95% in all cultures and pools were stored in vials containing 10% DMSO. To enrich for clones with high productivity, immunostaining of Rituximab on the plasma membrane (surface staining) was performed on the Rituximab expressing cell pool by staining for 30 min at 4 °C with 5 µg mL$^{-1}$ anti-human IgG antibody conjugated to FITC (Invitrogen, Carlsbad, CA, USA) and subsequent single-cell sorting using fluorescence activated cell sorting (FACS) and expansion. To select clones with the highest productivity, we used the titer-to-confluency method[44]. Confluency was determined on a Celigo cytometer (Nexcelom Bioscience, Lawrence, MA, USA) and titer was determined using biolayer interferometry. The number of clones screened by this method was 1000 (WT), 480 (6xKO), 1760 (11xKO), and 480 (14xKO).

**Quantification of secretome costs**. A genome-scale model of protein secretion was adapted for use here[24] using MATLAB 2018B [https://github.com/LewisLabUCSD/CHOSecretoryKO]. Briefly, this model was adapted to focus on the bioenergetic costs of protein synthesis and translocation of each host-cell secreted protein. Information on each secreted protein in CHO cells was obtained, including mRNA and protein sequence, signal peptides, etc. We then simulated the cost of producing each protein. Each cost was scaled by the measured mRNA levels using published RNA-Seq from CHO–S cells grown under similar conditions[25], and this was used as a proxy for the relative resources allocated to each secreted protein. Entrez gene identifiers were matched to their corresponding entry number in the UniProt database to determine presence of signal peptide. For those genes without a UniProt match, we used the online tool PrediSi[45] to determine the presence of a signal peptide. To estimate ATP cost of secretion for all secreted genes identified, we added the following costs, adapted from the genome-scale model of protein secretion[24]. First, the energy cost of protein translation was equal to 4x$L$ ATP molecules where $L$ is the length of the amino acid sequence. Next, the average cost of signal peptide degradation was equal to 22 ATP molecules. Finally, the energetic cost of translocation across ER membrane was equal to $L/40 + 2$. From this, we were able to quantify what proportion of all secreted protein was eliminated in our study.

**Identification of HCPs by mass spectrometry**. WT CHO–S was cultivated in duplicate in three separate DASGIP Mini bioreactor experiments. HCCF was obtained during late exponential phase by centrifugation at 500×$g$ for 10 min. Samples were TCA-precipitated with an overnight incubation in ice cold acetone, and the dry protein pellet was dissolved in 100 µl 8 M urea. Samples were treated first with 100 mM DTT (5 µl) and incubated at 37 °C for 45 min and then with 100 mM iodoacetamide (10 µl) and incubated in the dark for 45 min. Samples containing 100 µg of protein were digested overnight at 37 °C using trypsin after which 10% TFA was added, and samples were stage tipped. Data were acquired on Synapt G2 (Waters) Q-TOF instrument operated in positive mode using ESI with a NanoLock-spray source. During MS$^E$ acquisition, the mass spectrometer alternated between low and high-energy mode using a scan time of 0.8 s for each mode over a 50–2000-Da interval. Nanoscale LC separation of the tryptic digested samples was performed using a nanoAcquity system (Waters, Milford, MA, USA) equipped a nanoAcquity BEH130 C18 1.7 µm, 75 µm × 250 mm analytical reversed-phase column (Waters). A reversed-phase gradient was employed to separate peptides

using 5–40% acetonitrile in water over 90 min with a flow rate of 250 nL min$^{-1}$ and a column temperature of 35 °C. The data were analyzed using the Progenesis QI software v2.4 (NonLinear dynamics), which aligns the different runs ensuring that the precursor ions and identifications can be shared in between runs.

**HCP, antibody, and protein quantification**. HCP and antibody titers were quantified using biolayer interferometry on an Octet RED96 (Pall, Menlo Park, CA, USA). For HCP quantification, the anti-CHO HCP Detection Kit (Pall) was used according to the manufacturer's specifications. When needed, samples were diluted in sample diluent buffer before analysis. For Rituximab quantification, protein A biosensors (Pall) were equilibrated in PBS, and Rituximab was measured for 120 s at 30 °C. Absolute concentrations of Rituximab were calculated by comparison with a calibration curve generated from a dilution series of a human IgG control (31154, Thermo Fisher Scientific or A01006, Genscript, New Jersey, NJ, USA). When needed, samples were diluted in conditioned medium to fall within the range of the calibration curve. Regeneration of biosensor tips between measurements was performed in 10 mM glycine pH 1.7. The protein concentration of HCCF, cell extracts, and purified Rituximab was measured using a Nanodrop 2000 (Thermo Fisher Scientific). Before analysis, HCCF was TCA-precipitated, and whole cells were lysed using RIPA buffer. Rituximab was quantified using an extinction coefficient of 1.6 (A280 nm, 0.1% solution).

**Protein purification and glycan analysis**. Cell-free supernatants were prepared by centrifugation and directly loaded (100 mL) onto a 5-mL HiTrap MabSelect column (GE Healthcare). After washing the column with binding buffer (20 mM sodium phosphate, 0.15 M NaCl, pH 7.2), Rituximab was eluted with elution buffer (0.1 M sodium citrate, pH 3.5) into collection tubes containing 1 M TRIS-HCl, pH 9.0. The fractions containing Rituximab were pooled, diluted 10 times with 50 mM MES pH 6.0, and loaded onto a 5 mL HiTrap SP FF column (GE Healthcare). After washing the column with binding buffer (50 mM MES pH 6.0, 40 mM NaCl), Rituximab was eluted with elution buffer (50 mM MES pH 6.0, 200 mM NaCl). The fractions containing Rituximab were pooled, diluted 5 times with 20 mM TRIS pH 8.0, and loaded onto a 5-mL HiTrap Q FF column (GE Healthcare) equilibrated in buffer A (20 mM TRIS-HCl, 40 mM NaCl). The flow through containing Rituximab was collected and concentrated to 5 mg mL$^{-1}$. Samples for HCP and mAb titer analysis were collected after every chromatographic step. The final mAb samples were aliquoted, snap frozen in N$_2$ (l) and stored at −80 °C. N-linked glycans were released from purified Rituximab (12 µg) and fluorescently labeled with the GlycoWorks RapiFluor-MS N-Glycan Kit (Waters, Milford, MA, USA). Data were subsequently collected on a Thermo Ultimate 3000 UHPLC system equipped with a RS Fluorescence detector and coupled to a Thermo Fisher Velos pro iontrap MS run in positive mode. The separation gradient was 30–43% 50 mM ammonium formate buffer. The amount of N-glycan was determined by integrating the areas under the normalized fluorescence spectrum peaks with Thermo Xcalibur software v4.3 (Thermo Fisher Scientific) giving normalized, relative glycan quantities.

**Rituximab cell binding assay**. Rituximab was purified via a MabSelect column (GE Healthcare) and diluted to 5 µg mL$^{-1}$ using dilution buffer (PBS containing 3% BSA). These solutions were diluted further to respectively 2, 1, 0.5, 0.1, and 0.01 µg mL$^{-1}$ using dilution buffer. Ramos cells (2 × 10$^5$ cells per well; DSMZ, ACC 603) were blocked with PBS containing 5% BSA for 10 min and incubated with 100 µL of the Rituximab dilutions for 1 h at 4 °C. Cells were washed three times with PBS and subsequently incubated for 30 min at 4 °C with 5 µg mL$^{-1}$ phycoerythrin-conjugated goat F(ab′)2 anti-human IgG (Abcam, Cambridge, UK) as secondary antibody. After washing the cells 3x with PBS, binding was quantified in triplicate by flow cytometry using MACSQuant analyzer 10 VYB (Miltenyi Biotec, Bergisch Gladbach, Germany). Jurkat cells were used as CD20 negative cell line and a Rituximab biosimilar antibody (R&D systems, Minneapolis, MN, USA) was included as reference anti-CD20 antibody at the indicated dilutions. Analysis was performed using FlowJo v10 (BD Biosciences). A representative gating strategy is presented in Supplementary Fig. 7. Raw data and statistics are presented in Supplementary Data 3.

**RNA sample preparation and sequencing**. WT Rit, 6xKO Rit, 11xKO Rit cells (5 × 10$^6$) sampled from the DASGIP bioreactor on days 4, 6, and 8 were resuspended in RLT buffer (Qiagen, Hilden, Germany) containing 40-mM DTT. After storage at −80 °C, RNA was extracted using the RNeasy kit (Qiagen) followed by on-column DNAse digestion. RNA was eluted in 40 µl nuclease-free water, concentration was measured by Qubit (Thermo Fisher scientific) and the purity was checked on a Fragment Analyzer (Advanced Analytical). Complementary DNA synthesis was obtained from 1 µg of RNA using the High capacity cDNA RT kit (Thermo Fisher scientific). Samples were diluted to 60 ng µL$^{-1}$ in 50 µL and library preparation was performed with the TruSeq Stranded mRNA Library Prep Kit (Illumina, San Diego, CA, USA). Final RNA libraries were first quantified by Qubit and the size was checked on a Fragment Analyzer. Libraries were normalized to 10 nM and pooled and the final pool of libraries was run on the NextSeq platform with high output flow cell configuration (NextSeq® 500/550 High Output Kit v2, 300 cycles, Illumina). Raw data is deposited at the Gene Expression Omnibus and

Short Read Archive with accession number GSE144624 [https://www.ncbi.nlm.nih.gov/geo/query/acc.cgi?acc=GSE144624].

**RNA-Seq quantification and differential gene expression**. RNA-Seq quality was assessed using FastQC. Adapter sequences and low quality bases were trimmed using Trimmomatic[46]. Sequence alignment was performed using STAR[47] against the CHO genome (GCF_000419365.1_C_griseus_v1.0) with the default parameters. The expression of each gene was quantified using HTSeq[48]. We performed differential gene expression analysis using DESeq2[49]. After Benjamini-Hochberg FDR correction, genes with adjusted $p$-values < 0.05 and fold change >1.5 were considered as differentially expressed genes (DEGs).

**GSEA and enrichment strength analysis**. Gene set enrichment analysis (GSEA) was performed using the Broad Institute GSEA software[50]. A ranked list of genes (adjusted $p$-values < 0.05) using the differential expression values (fold change in the $\log_2$ scale) was run through the GSEA pre-ranked protocol. GSEA-pre-rank analysis was processed to detect significant molecular signature terms using MSigDB's Hallmark, Reactome, KEGG, and Gene Ontology Biological Process gene sets for the differential expressed genes. A molecular signature term was deemed significant if (1) after Benjamini–Hochberg false discovery correction, the molecular signature term has an adjusted $p$-value < 0.05; and (2) there are ≥10 genes presented in our gene list for the molecular signature term. The leading-edge analysis allows for the GSEA to determine which subsets (referred to as the leading-edge subset) of genes primarily contributed to the enrichment signal of a given gene set's leading-edge or core enrichment[50]. The leading-edge analysis is determined from the enrichment score (ES), which is defined as the maximum deviation from zero. The enrichment strength describes the strength of the leading-edge subset of a gene set (i.e., the interferon-alpha response in this study)[50]. Specifically, if the gene set is entirely within the first $N$ positions in the ranked differentially expressed gene list, then the signal strength is maximal or 100%. If the gene set is spread throughout the list, then the signal strength decreases towards 0%.

**Statistics and reproducibility**. Statistical parameters including the exact value of $n$, $p$ values, and the types of the statistical tests are reported in the figures and corresponding figure legends. Statistical analysis was carried out using Prism 8.4 (GraphPad Software). Statistical analysis was conducted on data from three or more biologically independent experimental replicates. Data distribution was assumed to be normal, but this was not formally tested. Comparisons between groups were planned before statistical testing and target effect sizes were not predetermined. Error bars displayed on graphs represent the mean ± SD of at least three independent experiments. Most experiments report technical replicates, whereas biological variability was addressed in the clonal variation experiment. *$p$ < 0.05, **$p$ < 0.01, ***$p$ < 0.001, and ****$p$ < 0.0001 were considered significant. The experiment that confirms the HCP-reduced phenotype in bioreactors was independently performed three times with similar results. No data was excluded from the study.

**Reporting summary**. Further information on research design is available in the Nature Research Reporting Summary linked to this article.

## Data availability
Raw RNA-Seq data that support the findings of this study have been deposited at the Gene Expression Omnibus and Short Read Archive with the accession number GSE144624 [https://www.ncbi.nlm.nih.gov/geo/query/acc.cgi?acc=GSE144624]. The authors declare that all other data supporting the findings of this study are available within the paper and its supplementary information files. The source data for Figs. 1–5, Table 1, and Supplementary Figs. 1–4 and 6 are provided as a source data file.

## Code availability
The code used for the analysis performed in this study is available at [https://github.com/LewisLabUCSD/CHOSecretoryKO].

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

## Acknowledgements

We would like to thank Helle Munck Petersen, Christian Oscar Wistrøm, Mikkel Schubert, Zulfiya Sukhova, Karen Kathrine Brøndum, Elham Maria Javidi, and Kristian Lund Jensen for support. This work was supported in part by the Novo Nordisk Foundation (NNF10CC1016517). N.E.L. and J.M.G. acknowledges support from NIGMS (R35 GM119850), a fellowship from the Government of Mexico (CONACYT), and the University of California Institute for Mexico and the United States (UC-MEXUS).

## Author contributions

B.O.P. and B.G.V. conceived the project. S.K., D.L., T.W., H.F.K., G.M.L., and N.E.L. designed the experiments. S.K., T.W., and L.E.P. selected knockout targets. T.W. performed proteomic analysis. S.K., J.A., M.D., and D.L. performed cell line cultivation, selection, and analysis. S.K. and S.S. performed protein purification and analysis. S.S. and T.L.J. performed binding assays and A.H.H. performed glycosylation analysis. N.E.L., J.M.G., A.W.T.C., and H.O.M performed in silico and RNA-Seq analysis. S.K. and N.E.L. wrote and edited the manuscript.

## Competing interests

A patent based on this work has been filed with authors B.O.P, B.G.V., and L.E.P. as inventor. The International Patent Application No. is EP20160166789. All other authors declare no competing interests.
