## [Peer Review File · Nature Communications]

Reviewers' Comments:

Reviewer #1:

Remarks to the Author:

Thank you for giving me the opportunity to review this manuscript by Kol et al. This is a very interesting, well designed and comprehensive study that will undoubtedly be of high interest to the bioprocessing community. I only have three questions for clarification and comment by the authors:

On page 11, line 312, the authors introduce a comparison based on specific HCP productivity. I suspect that this refers to secreted HCPs only - can the authors please clarify in the text.

On page 14, line 373, the authors mention that the bioreactor experiments were carried out, in part, to ensure that the reduced HCP content was retained. In Industry this usually involves stability studies over multiple generations. Can the authors please clarify if any such stability study was carried out and if not, over how many generations stability was assessed.

In a broader context and in relation to the use of model-guided genetic engineering, can the authors comment on whether the model simulation results for the KO cell lines were numerically accurate. In other words, can the model predict the % reduction of secreted HCP content for these cell lines with any accuracy or can it only be used for screening at this stage?

Reviewer #2:

Remarks to the Author:

Dear Editor,

Thank you very much for the opportunity of reviewing the manuscript entitled: "Multiplex secretome engineering enhances recombinant protein production and purity" by Kol et al. The manuscript describes the generation via a multiplexed CRISPR/Cas9 knock-out strategy of CHO lines that very significantly reduce the production of abundant host cell protein (HSCPs). These knock-out clones surprisingly have a superior fitness while producing high titer of antibodies in a reduced HSCP secretory background that facilitates the removal of unwanted impurities. Overall, the manuscript shows the feasibility and validation to larger multi-gene scale of an idea previously shown for single knock-out genes. The data are logically presented, the conclusions are supported by the data and the manuscript is clearly written.

Few revisions will further improve the quality of the work:

- The data from Fig. 5 appear to be originated from one single high-producing rituximab clone for each the WT, 6xKO and 11xKo lines. At least two additional clones for each genetic condition should be tested to show reproducibility of the same findings in multiple clones.
- The conclusion on line 451: "Importantly, the ability to generate high mAb producers was not perturbed" is not accurate as rituximab-producing clones from the 14xKO were not obtained, and the clone from the 11xKO seems to express lower titer levels than WT or 6xKO (Fig. 5B). Actually the Authors should comment on possible reasons why clones from the 14xKO were not obtained.
- The panel Figure S2 (transfection efficiency) does not have standard deviation. Repeated experiments should be provided with statistical analysis.

Reviewer #1:

Thank you for giving me the opportunity to review this manuscript by Kol et al. This is a very interesting, well designed and comprehensive study that will undoubtedly be of high interest to the bioprocessing community. I only have three questions for clarification and comment by the authors:

1. On page 11, line 312, the authors introduce a comparison based on specific HCP productivity. I suspect that this refers to secreted HCPs only - can the authors please clarify in the text.

The term HCP refers to proteins actively secreted or passively released from dead cells throughout the article. The specific HCP productivity provides a measure that takes into account differences in cell density and it refers to all HCPs, not just the secreted ones.

2. On page 14, line 373, the authors mention that the bioreactor experiments were carried out, in part, to ensure that the reduced HCP content was retained. In Industry this usually involves stability studies over multiple generations. Can the authors please clarify if any such stability study was carried out and if not, over how many generations stability was assessed.

The knockout clones that were used in the bioreactor experiments have spent 67 days (6xKO) and 56 days (11xKO) in culture from thawing the original clones until Rituximab producing clones were banked. The doubling time in shake flasks is approximately 21 hours. This means that the HCP-reduced phenotype is stable over at least 77 (6xKO) and 64 (11xKO) generations. Comments addressing this have been added to the results and discussion sections.

3. In a broader context and in relation to the use of model-guided genetic engineering, can the authors comment on whether the model simulation results for the KO cell lines were numerically accurate. In other words, can the model predict the % reduction of secreted HCP content for these cell lines with any accuracy or can it only be used for screening at this stage?

We have clarified in the conclusion that the modeling tool aids in the identification of more costly HCPs.

Reviewer #2:

Thank you very much for the opportunity of reviewing the manuscript entitled: "Multiplex secretome engineering enhances recombinant protein production and purity" by Kol et al.

The manuscript describes the generation via a multiplexed CRISPR/Cas9 knock-out strategy of CHO lines that very significantly reduce the production of abundant host cell protein (HSCPs).

These knock-out clones surprisingly have a superior fitness while producing high titer of antibodies in a reduced HSCP secretory background that facilitates the removal of unwanted impurities. Overall, the manuscript shows the feasibility and validation to larger multi-gene scale of an idea previously shown for single knock-out genes. The data are logically presented, the conclusions are supported by the data and the manuscript is clearly written.

Few revisions will further improve the quality of the work:

1. The data from Fig. 5 appear to be originated from one single high-producing rituximab clone for each the WT, 6xKO and 11xKo lines. At least two additional clones for each genetic condition should be tested to show reproducibility of the same findings in multiple clones.

We have tested two additional clones for every genetic condition, as well as one additional wild-type clone. The HCP-reduced phenotype was observed again. Results are summarized in Fig. S4. Comments have been added to the results and discussion sections.

2. The conclusion on line 451: “Importantly, the ability to generate high mAb producers was not perturbed” is not accurate as rituximab-producing clones from the 14xKO were not obtained, and the clone from the 11xKO seems to express lower titer levels than WT or 6xKO (Fig. 5B). Actually the Authors should comment on possible reasons why clones from the 14xKO were not obtained.

The text in the conclusion has been changed to: “Importantly, the ability to generate high mAb producers was not perturbed in the 6xKO and 11xKO cell lines”. The reason that high mAb-producing clones from the 14xKO cell line were not obtained remains unclear at present. Our efforts were hampered by the fact the single cell survival rate was decreased considerably in the 14xKO cell line. We therefore needed to seed a very large number of 384-well plates, which subsequently had to be collapsed into 96-well plates for mAb productivity screening. We did not have the resources to perform these experiments. As HCPs were not reduced further in this knockout, we decided to omit the 14xKO cell line from the study.

3. The panel Figure S2 (transfection efficiency) does not have standard deviation. Repeated experiments should be provided with statistical analysis.

The transfection efficiency analysis has indeed only been analyzed once. It was a minor step in the process to clone selection and the measurement was only important to show consistency among the transfections to verify there were no large phenotypic differences in the different pools made, which might impact clone phenotypes. Since the transfection levels were consistent, we just moved on to clone selection.

Reviewers' Comments:

Reviewer #1:

Remarks to the Author:

The revised manuscript is improved in terms of clarity and accuracy. I particularly appreciate the acknowledgment of limitations of the experimental design/study as well as the additional work that has been carried out to provide evidence of reproducibility. I have no further comments for the authors.

Reviewer #2:

Remarks to the Author:

The Authors have satisfactorily addressed the concerns.

Kol et al.

NCOMMS-19-16311A

REVIEWERS' COMMENTS:

Reviewer #1 (Remarks to the Author):

The revised manuscript is improved in terms of clarity and accuracy. I particularly appreciate the acknowledgment of limitations of the experimental design/study as well as the additional work that has been carried out to provide evidence of reproducibility. I have no further comments for the authors.

Reviewer #2 (Remarks to the Author):

The Authors have satisfactory addressed the concerns.

RESPONSE

No response required.